# D-dimer Testing in Pulmonary Embolism with a Focus on Potential Pitfalls: A Narrative Review

**DOI:** 10.3390/diagnostics12112770

**Published:** 2022-11-12

**Authors:** Loris Wauthier, Julien Favresse, Michaël Hardy, Jonathan Douxfils, Grégoire Le Gal, Pierre-Marie Roy, Nick van Es, Cihan Ay, Hugo ten Cate, Thierry Vander Borght, Michaël V. Dupont, Thomas Lecompte, Giuseppe Lippi, François Mullier

**Affiliations:** 1Department of Laboratory Medicine, Clinique St-Luc Bouge, 5004 Namur, Belgium; 2Department of Pharmacy, Namur Research Institute for Life Sciences, University of Namur, 5000 Namur, Belgium; 3Namur Thrombosis and Hemostasis Center (NTHC), CHU UCL Namur, Université Catholique de Louvain, 5530 Yvoir, Belgium; 4Department of Pharmacy, Namur Thrombosis and Hemostasis Center (NTHC), Namur Research Institute for Life Sciences (NARILIS), University of Namur, 5000 Namur, Belgium; 5QUALIblood s.a., 5000 Namur, Belgium; 6Department of Medicine, University of Ottawa, Ottawa, ON K1N 6N5, Canada; 7Department of Emergency Medicine, CHU Angers, Institut MITOVASC, Equipe CARME, UMR CNRS 6015-INSERM 1083, UNIV Angers, F-CRIN INNOVTE, 49000 Angers, France; 8Amsterdam Cardiovascular Sciences, Pulmonary Hypertension & Thrombosis, Amsterdam UMC Location Universiteit van Amsterdam, 1105 Amsterdam, The Netherlands; 9Department of Vascular Medicine, Amsterdam UMC Location Universiteit van Amsterdam, 1105 Amsterdam, The Netherlands; 10Clinicial Division of Haematology and Haemostaseology, Department of Medicine I, Medical University of Vienna, 1090 Vienna, Austria; 11Department of Internal Medicine and Thrombosis Expertise Center, CARIM School for Cardiovascular Diseases, Maastricht University Medical Center, 6229 HX Maastricht, The Netherlands; 12Center for Thrombosis and Hemostasis, Gutenberg University Medical Center, 55131 Mainz, Germany; 13Department of Nuclear Medicine, CHU UCL Namur, 5530 Yvoir, Belgium; 14Institute of Experimental and Clinical Research, UCLouvain, 1348 Louvain-la-Neuve, Belgium; 15Namur Research Institute for Life Sciences, University of Namur, 5000 Namur, Belgium; 16Department of Radiology, CHU UCL Namur, 5530 Yvoir, Belgium; 17Division of Vascular Medicine, CHRU de Nancy, Université de Lorraine, 54000 Nancy, France; 18Section of Clinical Biochemistry, University Hospital of Verona, 37134 Verona, Italy

**Keywords:** D-dimer, venous thromboembolism, pulmonary embolism, pre-analytical, analytical, post-analytical, In Vitro Diagnostic Regulation, ISO15189

## Abstract

D-dimer is a multifaceted biomarker of concomitant activation of coagulation and fibrinolysis, which is routinely used for ruling out pulmonary embolism (PE) and/or deep vein thrombosis (DVT) combined with a clinical pretest probability assessment. The intended use of the tests depends largely on the assay used, and local guidance should be applied. D-dimer testing may suffer from diagnostic errors occurring throughout the pre-analytical, analytical, and post-analytical phases of the testing process. This review aims to provide an overview of D-dimer testing and its value in diagnosing PE and discusses the variables that may impact the quality of its laboratory assessment.

## 1. Introduction

What is universally referred to as “D-dimer” is actually a mixture of fibrin degradation products (FnDP) that contain a “D-dimer” motif, obtained from cross-linked fibrin digestion by plasmin [1,2,3,4]. Thrombin, activated factor XIII (factor XIIIa), and plasmin are the three successive enzymes responsible for D-dimer formation. Fibrinogen contains a central E-domain connected to two outer D-domains and consists of three pairs of polypeptide chains (Aα-, Bβ-, and γ-) [5,6]. Thrombin releases fibrinopeptides A and B, generating fibrin monomers and unraveling two cryptic polymerization sites located on the E domain. Self-assembling of fibrin monomers leads to the formation of a network, which becomes insoluble (fibrin deposits) after reaching a critical size [1]. Thrombin activates FXIII into FXIIIa in a reaction that is enhanced by fibrin [7]. FXIIIa in turn promotes the generation of stable fibrin clots, the lysis of which is slowed down [1,4]. The generation of plasmin from plasminogen catalyzed by tissue plasminogen activator and urokinase-type plasminogen activator leads to the degradation of fibrin by cleavages at several specific sites [4]. Fibrinolysis generates products with a wide range of molecular masses, referred to as FnDP. The smallest final digestion product of fibrin clots is the D-dimer/fragment E (DD/E) complex, which presents two adjacent covalently bound D-domains from two fibrin monomers cross-linked by factor XIIIa [4]. 

Initially, the term “D-dimer fragment” only referred to the DD/E complex [1,8]. Nonetheless, the range of species recognized by antibodies in D-dimer assays is wider, including species with masses ranging from approximately 200 to over 10,000 kDa [1,9]. In healthy subjects, D-dimer is detectable in small amounts because of the physiological conversion of fibrinogen to fibrin [4,10]. The half-life of D-dimer in vivo is estimated to be approximately 8 h [11]. 

D-dimer immunoassays are designed to detect a specific epitope on degradation products of factor XIIIa-cross-linked fibrin and should therefore not recognize X, Y, D, or E fragments but only the D-dimer motif in the smallest DD/E fragment and in larger FnDP [3]. 

## 2. Interest in D-dimer Testing in PE

Pulmonary embolism (PE) is defined as the obstruction of blood flow in a pulmonary artery by an embolus that has usually originated from a thrombus in the lower limb veins. Its incidence is approximately 60 to 120 per 100,000 people/year. The clinical picture mainly includes acute chest pain, shortness of breath (i.e., dyspnea), or syncope, and diagnosis requires chest imaging [12]. D-dimer testing is widely used as a gold standard for ruling out venous thromboembolism (VTE) in patients classified as having a low-intermediate pre-test probability based on a clinical decision rule (CDR) or implicit assessment [1,4,13,14,15,16]. In addition to almost all cases of VTE, increased D-dimer levels may also be observed in many conditions, such as infection, pregnancy, trauma, cancer, and aging as well as in hematomas or interstitial hemorrhages [2]. Interpretation of D-dimer assays should be performed following local regulations that may apply, as some uses may fall outside the requirements of intended use approved by the local regulatory agency [17,18,19]. The use of tests outside those directives (e.g., age-adjusted cut-offs) may constitute a laboratory-developed test. Other indications for D-dimer testing include assessing the risk of recurrent thrombosis and guiding anticoagulant therapy and diagnosing and monitoring disseminated intravascular coagulation although the use of D-dimer in those settings is controversial or less well-established [1,4,20]. Table 1 summarizes the characteristics of some relevant central laboratory and point-of-care (POC) assays, including the method type and the “intended use” issued by the U.S. Food and Drugs Administration (FDA) [18]. Regarding test performance, the recommendations from the FDA are ≥95% (with a lower limit of confidence interval (CI) ≥90%) sensitivity and ≥97% (with a lower limit of CI ≥95%) negative predictive value (NPV) [21], while the Clinical and Laboratory Standards Institute (CLSI, former NCCLS) suggests specifications of ≥97% (with a lower limit of CI ≥90%) sensitivity and ≥98% (with a lower limit of CI ≥95%) NPV to rule out VTE. Quantitative assays have higher sensitivity than qualitative or semi-quantitative assays and should hence be preferred [3]. Of note, accurate estimation of the performance of D-dimer immunoassays is pivotal and required by the In Vitro Diagnostics Regulation (IVDR) [22]. Regarding POC assays, whole-blood D-dimer assays may display higher specificity for PE than laboratory assays (i.e., 69%) but at the expense of lower sensitivity (i.e., 87%) [23,24]. 

The use of validated CDR in combination with D-dimer testing is recommended to identify patients with suspected VTE in whom the diagnosis can be ruled out without imaging [2,4,13,25,26,27,28,29]. A score is assigned to an individual based on signs, symptoms, and risk factors, which can be used to classify patients into risk categories for VTE (e.g., low, intermediate, or high probability). Straightforward two-level scores are preferred since they perform as well as three-level scores and are easier to use [25,26]. Numerous CDR for suspected PE are available, among which the Wells’ PE and revised Geneva scores are the most widely used and display comparable diagnostic performances [1,16,30,31,32,33]. By using an approach that combines a CDR with D-dimer testing to rule out PE, the risk of withholding imaging in patients with false-negative D-dimer (failure rate) is very low [1,2,4,13,34]. The use of a combination of CDR and D-dimer testing may be safely applied when the incidence of thromboembolic events is below 1–2% after 3 months in patients in whom VTE was excluded without imaging testing [35,36]. Imaging testing is recommended, and D-dimer testing is unnecessary in patients with “high” or “likely” probability [2,36,37]. 

The use of D-dimer testing alone was evaluated with the YEARS algorithm that, in essence, excluded PE in all patients with a high-sensitive D-dimer < 500 µg/L fibrinogen-equivalent units (FEU) [33]. In the YEARS study, the overall failure rate (1 minus NPV) of this diagnostic algorithm was well below the accepted threshold of 2%. The recently introduced PEGeD algorithm, which also uses varying D-dimer thresholds dependent on the clinical pretest probability, maintains the approach of not using D-dimer alone to rule out PE in patients with a high pretest probability based on the Wells’ score [38].

An age-adjusted cut-off for D-dimer is defined as age times 10 µg/L in those aged 51 years or older, while the traditional threshold of 500 µg/L FEU is used in those 50 years or younger. Age-adjusted cut-offs combined with results of a clinical score (i.e., Wells’ or revised Geneva score) were demonstrated by the ADJUST-PE study to enable safe PE exclusion without additional computed tomography pulmonary angiography (CTPA) [27,39]. The use of such cut-offs is now one of the recommended diagnostic approaches for patients with suspected PE in order to enhance the clinical specificity while maintaining a clinically usable NPV [13,30,40,41,42,43]. For example, the European Society of Cardiology recently endorsed the use of age-adjusted cut-offs for ruling out PE [28]. 

Interestingly, D-dimer may also be used to rule out VTE in specific clinical circumstances such as pregnancy, cancer, or kidney disease [11].

## 3. Pre-Analytical Step

Among all the various steps of the total testing process, the pre-analytical phase is currently considered as the main source of diagnostic errors, accounting for 60–70% of all laboratory mistakes [44]. The International Organization for Standardization 15189:2012 standard for laboratory accreditation is defined as “processes that start, in chronological order, from the clinician’s request and include the examination request, preparation and identification of the patient, collection of the primary sample(s), the transport to and within the laboratory, and end when the analytical examination begins” [45]. The pre-analytical requirements for D-dimer analysis are often wrongly assimilated to those for other coagulation tests [46,47]. This misperception may come from the fact that, although testing methods are diametrically different, D-dimer and other hemostasis tests are generally performed using the same citrated tube. 

The first variable that affects the D-dimer measurement is sample collection. To perform venipuncture, it is recommended that the tourniquet should not obstruct the arterial blood flow, and venous blood flow should be freed as soon as the needle enters the vein or when the first tube starts to fill [48,49]. If applied longer than 1 to 2 min, the obstruction of blood flow may induce both hemoconcentration and thrombin generation, therefore interfering with accurate hemostasis testing [48,49,50]. In a study on the effect of tourniquet application on D-dimer, 1 min of venous stasis already increased D-dimer values on average by 7.9% using the Vidas DD assay (BioMérieux, Marcy l’Etoile, France) [50]. 

The act of collection should be performed by ordinary straight needles with a caliber between 19–22 gauge (G) and confer as little trauma as possible [48]. Butterfly devices, even with small-sized needles, constitute an appropriate alternative approach to standard straight needles when needed [50,51] although a mandatory discard tube must be drawn prior to sample collection to ensure removing any air present within the tubing. The presence of air may be responsible for inadequate volume in the tube and subsequent disturbances or interferences [52]. Straight-needle venipunctures are to be preferred when other hemostasis analyses are requested [13]. In this case, the use of a discard tube does not have a significant impact on D-dimer results and is therefore unnecessary when D-dimer is the only test requested [53].

The tube material as well as the additive used for anticoagulation may also be sources of variation. Different tube materials were studied (i.e., glass or polyethylene terephthalate plastic collection tubes (Vacutainer, Becton Dickinson (BD), Franklin Lakes, NJ, USA) as well as glass citrated collection tubes (Vacuette, Grenier bio-one, Germany)) and showed no significant differences in D-dimer values [54,55,56]. Altogether, the choice of silicone-coated glass or polypropylene plastic tubes does not substantially impact the quality of D-dimer measurement. The CLSI and the World Health Organization recommend the use of collection tubes containing 3.2% (105–109 mmol/L) buffered sodium citrate anticoagulant [49,52]. Alternative sample matrices should be locally validated before being used in routine clinical practice. Few D-dimer assays (including POC tests) also allow using heparinized or ethylenediamine tetraacetic acid plasma, whilst only citrate anticoagulated blood is recommended for others [57]. Reports in the literature do not agree on the influence of heparin tubes, stating that a modest bias could appear, which could be corrected by a correction factor to adjust for the dilution induced by the presence of the anticoagulant [58,59,60]. Importantly, a blood-to-anticoagulant ratio of 9:1 must be respected once the specimen is collected to prevent underestimation of D-dimer due to excessive sample dilution [4,49,61]. Sample collection should ensure free flow into the tube and be mixed within 30 s after completing the phlebotomy (i.e., three to six complete inversions), thus ensuring homogenization of blood and anticoagulant [49]. 

Samples delivery should be performed as fast as possible (i.e., usually <1 h) after collection, in a vertical position and at ambient temperature (15–22 °C) [48,49,52]. Owing to accessibility and rapidity, a reduction in turnaround time (TAT) may be obtained using a pneumatic tube system (PTS). Although a low impact of PTS on D-dimer test results has been noted [58,62,63], PTS should be assessed and validated locally since these systems are rather heterogeneous in terms of length, materials, internal diameter, acceleration (or deceleration) forces, and speed [62,64].

Sample processing is another important pre-analytical variable. Once received in the laboratory and prior to analysis, the specimens should be carefully examined to identify those that do not comply with pre-analytical requirements (e.g., inappropriate additive, identification errors, insufficient volume, presence of clots). To ensure patient safety, noncompliant samples should be rejected, or test results should be otherwise suppressed [48]. Automation (i.e., pre-analytical modules) may help identify pre-analytical issues [65]. Then, prior to performing most D-dimer analysis, plasma needs to be separated from cellular components, usually through centrifugation at 1500× *g* for at least 15 min at ambient temperature [52]. Shorter centrifugation schemes may also be used, especially for stat testing, and are of major interest to ensure shorter TAT [66,67]. The impact of the temperature of centrifugation was also shown to be non-significant [34,68].

Finally, the stability of D-dimer in collecting tubes prior to analysis is of the utmost importance to ensure appropriate results. Various real-life studies suggested that sample stability might be sufficient to allow longer delays than stated by current recommendations (i.e., no more than 4 h room temperature (15–25 °C) prior to D-dimer testing [48]). D-dimer is stable for at least 24 h (in plasma/whole blood) at room temperature or at 2–8 °C with many different D-dimer immunoassays before performing D-dimer testing. Several freezing-thawing cycles did not induce significant changes in D-dimer values [58,69,70,71]. Due to the important clinical decisions that could be taken (e.g., ruling out acute episodes of PE), rapid D-dimer testing is typically required, and delayed analysis (e.g., frozen samples) would prove most useful in research settings [13,69,72,73,74,75]. 

Importantly, like all immunoassays, D-dimer testing may be affected by interferences that significantly impact test results (i.e., 0.4 to 4.0% of all analyses) [65]. With a prevalence ranging between 30–70% of all unsuitable specimens, in vitro hemolysis is the most frequent interference in hemostasis testing and is caused by various conditions (e.g., hemolytic anemia, metabolic disorders), traumatic venipuncture, sample transportation, processing (e.g., delay before centrifugation), and storage (e.g., unsuitable conditions of temperature and duration) [76,77,78,79,80]. Notably, it has been observed that the majority of hemolyzed samples (±95%) in clinical laboratories are only “mildly” hemolytic (e.g., cell-free hemoglobin concentration between 0.3–0.6 g/L, giving a faint reddish hue to the sample) [76,78,81]. D-dimer values might still be reliable when the concentration of cell-free hemoglobin remains within a non-interference limit (i.e., <3 g/L) and could hence be safely reported to the clinicians [78]. Greater hemolysis might, however, lead to biased results, which would need to be suppressed in order to safeguard patient safety [77,78,82,83]. The impact of other frequent sources of interference, such as lipemia, icterus, proteinemia (including monoclonal gammopathy) and heterophilic antibodies, is less-documented [77,84,85,86]. Some studies observed no impact of lipemia and icterus [87,88,89,90,91,92,93]. Spurious D-dimer results were only observed with the STA reagents (Diagnostica Stago, Asnières sur Seine, France) when adding high concentrations of free bilirubin at 80 mg/dL and above [94]. Moreover, the HIL check was able to detect lipemic samples (triglyceride levels > 400 mg/dL) on the CS-5100 D-dimer assay (Sysmex, Kobe, Japan) in three out of four samples [95]. Recently, patient pools with low D-dimer concentrations (<750 µg/L FEU) were shown to be significantly affected by lipemia (HemosIL HS500, Werfen, San Diego, CA, USA) [96]. Interestingly, excessive lipemia might be decreased using high-speed centrifugation (e.g., at 10,000 *g*) [97]. Multiple case reports suggested or illustrated an interference of monoclonal gammopathy in D-dimer assays [85,98,99,100]. Few reports on heterophilic antibodies are available on interference with D-dimer assays [84,101,102,103,104]. Recently, such cases were also reported in COVID-19 patients [105]. Such interferences may be detected and prevented using commercially available heterophilic blocking agents [84,85] or by using a second (and unaffected) assay [65,85]. Research and identification of interferences may be performed based on a practical algorithm [65]. 

## 4. Analytical Step

The first method developed for D-dimer testing was a microplate enzyme-linked immunosorbent assay (ELISA) [1]. The principle relies on the use of D-dimer-specific capture antibodies fixed on a plate, followed by the generation of a colorimetric detection when a labeled antibody is added and binds to the immobilized D-dimer antigen (sandwich assay) [1,2]. These techniques were labor-intensive and had poor analytical precision and reproducibility [2,106]. The following generations of assays were mainly based on qualitative agglutination of antibody-coated latex microparticles detected by visual inspection [1,3,4,107]. Latex-enhanced immunoturbidimetric assays today reach a sensitivity close to ELISAs but offer significantly shorter TAT [3]. 

The automated Vidas enzyme-linked immunofluorescence assays (ELFA) were shown many years ago to display comparable analytical performance to microplate ELISA [3,87]. The Vidas method was thoroughly clinically validated [34,57,108,109,110,111] and is often considered the reference quantitative immunoassay for D-dimer testing [110,112,113]. Finally, chemiluminescent enzyme immunometric assays (CLIA) use magnetic particles coated with D-dimer-specific monoclonal antibodies, while detection is enabled by ulterior fixation anti-D-dimer antibodies labeled with chemiluminescent substances that generate a detectable light signal [3]. The sensitivity of CLIA is similar to or even better than ELISAs, latex-enhanced immunoturbidimetric, and ELFA assays [3,59,112]. 

Notably, POC D-dimer assays using methods involving monoclonal antibodies may facilitate the triage of patients suspected of VTE [3,114,115,116]. Accessibility to general practitioners shortens TAT [23,114,117], and the use of whole-blood samples enables testing without centrifugation [3,114]. Method design may be qualitative or (semi-)quantitative [3,118]. Only POC tests validated in clinical trials and cleared by the FDA, the European Community, or other similar regulatory agencies should be used for excluding PE. Table 1 recapitulates relevant central laboratory and POC D-dimer assays and their characteristics and intended use as issued by the FDA. 

D-dimer assays may also vary importantly between laboratories. In a study on 353 laboratories using the seven most frequently used D-dimer immunoassays, the authors observed that different assays could produce high variations in D-dimer concentrations between assays, causing misclassifications around the 500 ng/mL FEU cut-off [119]. An inter-method coefficient of variation of 42% was, furthermore, observed by the Coagulation Resource Committee of the College of American Pathologists in a survey involving 3800 laboratories [120].

Importantly, D-dimer units vary according to the type of calibration material [9,121,122]. For FEU calibrators preparation, plasmin degradation of purified fibrinogen clotted in the presence of factor XIII is used, whilst calibrators are composed of purified D-dimer for D-dimer units (DDU), respectively [4]. Based on their molecular mass, DDU (195 kD) could be converted to FEU (340 kD) by applying a factor of correction (i.e., approximately 2×) [2,3,4,9,13].

Given the coexistence of multiple sources of heterogeneity in D-dimer assays (e.g., specificity of proprietary monoclonal antibodies, the existence of various fragments obtained from the digestion of cross-linked fibrin from low-molecular-mass forms to high-molecular-mass forms, the lack of certified IQC or calibrators, different reporting units and clinical cut-offs), the development of an international reference for calibration and standardization of D-dimer immunoassays are still unmet needs unlikely to be achieved [9,120,123,124,125,126,127,128,129,130,131,132]. Nonetheless, efforts are underway to provide more comparable results through the use of mathematical models and conversion factors [119,121,122,127,133]. Although such efforts towards harmonization have been made, a single diagnostic cut-off does not seem attainable since false-negative test results would have a harmful clinical impact [9]. According to the Fibrinolysis and Disseminated Intravascular Coagulation Scientific Standardization Subcommittees of the International Society on Hemostasis and Thrombosis, a consensus reference for D-dimer immunoassays might be obtained from a stable freeze-dried reference material containing high concentrations of D-dimer of heterogeneous species [132]. A study among the latest attempted to produce such material but found major instability due to structural rearrangements and amyloid formation of FnDP [134]. Another approach would be to selectively target low- and middle-molecular-mass FnDP species and higher-molecular-mass forms, using adequate monoclonal antibodies [126].

Prospective studies were carried out to validate cut-offs with some reagents (e.g., Vidas, AxSYM, STA Liatest) [1,4,34,108,109,110,129,135] but not with others. When validation studies are missing, a comparison with validated assays must be performed. This will also be required under the new IVDR [22]. It is the role of manufacturers to stay abreast of the most recent literature regarding their assays and suggest revised cut-offs when needed [136].

Recommendations stated that a coefficient of variation <10% should be observed around the diagnostic decision cut-off, and linearity should extend from 50 to 5000 µg/L FEU [132]. The CLSI has proposed to aim at a precision ≤7.5% around the assay-specific diagnostic threshold [17]. Importantly, there should be no cross-reaction with fibrinogen or fibrinogen-degradation products and preferably not with fibrin and fibrinogen fragments released from proteolysis mediated by various enzymes other than plasmin [132]. 

## 5. Post-Analytical Step

The International Organization for Standardization 15189:2012 standard defines the post-analytical phase as “processes following the examination including systematic review, formatting and interpretation, authorization for release, reporting and transmission of the results, and storage of samples of the examinations” [45]. The post-analytical phase is still the cause of a large number of diagnostic errors (15–20%), which may jeopardize patient safety [125]. 

Adequate reporting of D-dimer results is crucial. Several different measure units (i.e., ng/mL, μg/L, mg/L, g/L, μg/mL, mg/mL, and g/dL) are used [3] to report results obtained with diverse assays [2,137,138]. The coexistence of FEU, DDU, and different measure units is a major issue that may cause confusion among clinicians and potentially lead to misinterpretation of test results [3,4,125]. A total of 14 combinations of units and calibrations may hence be used, and surveys reveal an important variability in D-dimer test reporting between laboratories [9,120,125,139]. It is now recommended to favor “µg/L” (or “ng/mL”), which is the unit that best approximates the International System [13,125]. Additionally, the possibility to use age-adjusted cut-offs (or clinical probability-adjusted cut-offs) is another source of complexity for reporting D-dimer test results [125]. 

Recently, the SARS-CoV-2 pandemic and associated thrombotic events led to a considerable increase in D-dimer measurements as well as in the number of publications on this subject, with many presenting D-dimer reporting issues, ultimately generating confusion and misinterpretations (e.g., absence of units, identification and type of the assay used, age-specific cut-offs) [140]. Overcoming these challenges in reporting D-dimer through standardization is crucial [120,128,132]. 

Finally, to ensure the clinical usefulness of D-dimer in urgent situations, TAT offered by clinical laboratories must be the shortest possible. An acceptable TAT < 1 h has been proposed, which seems appropriate given the urgent (clinical) nature of this test [13]. A shorter TAT may be obtained by selecting assays that guarantee a wide range of linearity (i.e., up to 5000 µg/L FEU), therefore avoiding additional dilutions, ensuring faster pre-analytical steps (e.g., use of PTS and high-speed centrifugation), and using bedside POC analyzers [3,13,58,62,63,66,117,129]. Current methods (e.g., ELFA, CLIA) require an analytical processing time ranging between 15–40 min. In a European study, 81% of participant laboratories responded that D-dimer testing was available 24/7 [14].

## 6. Conclusions

D-dimer is a multifaceted biomarker of activation of coagulation and fibrinolysis and is the main laboratory test for patients with suspected VTE (PE and/or DVT). It is currently recommended to use D-dimer assays with high sensitivity to maximize the NPV (i.e., ≥95% sensitivity and ≥97% NPV). In patients with a low or intermediate clinical pretest probability based on structured CDR, a normal D-dimer test result safely rules out acute PE and the need for additional imaging, whilst values above the cut-off should trigger further testing to confirm or refute the diagnosis. The use of age-adjusted cut-offs and, more recently, clinical probability-adjusted cut-offs have been introduced to increase the diagnostic specificity. Pre-analytical requirements for D-dimer differ from other hemostasis tests, as this biomarker appears to be less sensitive to some pre-analytical variables. D-dimer suffers from high inter-laboratory variation, and results obtained from a particular assay cannot be simply extrapolated to another. Reporting of D-dimer is complex, and clinicians should be aware of the possible use of different measuring units. Further studies are needed to promote harmonization of D-dimer measurement and reporting. Systematic reviews on narrower topics on the use of D-dimer in PE are also needed. Future regulations such as the coming IVDR in Europe will have a major impact on the use of D-dimer for PE exclusion and will bring a better definition of the intended use of each D-dimer assay.

## Figures and Tables

**Table 1 diagnostics-12-02770-t001:** Characteristics of common D-dimer assays. Relevant central laboratory and point-of-care assays are shown along with the manufacturer, method design, method principle, unit type, cut-off mentioned by the manufacturer, and the intended use issued by the FDA regarding PE. DDU, D-dimer units; FDA, U.S. Food and Drugs Administration; FEU, fibrinogen-equivalent units; NA, not applicable; PE, pulmonary embolism; POC, point-of-care.

Assay Type	Assay Name	Manufacturer	Design	Principle	Unit Type	Manufacturer’s Cut-Off	FDA Intended Use (PE)
Central lab	Advanced D-Dimer	Siemens Healthcare Diagnostics (previously Dade Berhing)	Quantitative	Latex-enhanced turbidimetric immunoassay	FEU	BCS System: 1.6 mg/L Sysmex CA-1500: 1.0 mg/L	Aid in diagnosis
Central lab	AxSYM D-dimer	Abbott laboratories	Quantitative	Enzyme-linked fluorescent assay	FEU	500 µg/L	NA
Central lab	Diazyme D-Dimer	Diazyme Laboratories	Quantitative	Latex-enhanced turbidimetric immunoassay	FEU	0.5 µg/mL	Aid in diagnosis
Central lab	HemosIL AcuStar D-Dimer	Werfen (previously Instrumentation Laboratory)	Quantitative	Chemiluminescent immunoassay	FEU	500 µg/L	Aid in diagnosis
Central lab	HemosIL D-Dimer (±HS)	Werfen (previously Instrumentation Laboratory)	Quantitative	Latex-enhanced turbidimetric immunoassay	DDU	230 µg/L	Exclusion
Central lab	HemosIL D-Dimer HS 500	Werfen (previously Instrumentation Laboratory)	Quantitative	Latex-enhanced turbidimetric immunoassay	FEU	500 µg/L	Exclusion
Central lab	Innovance D-Dimer	Siemens Healthcare Diagnostics	Quantitative	Latex-enhanced turbidimetric immunoassay	FEU	0.5 mg/L	Exclusion
Central lab	STA Liatest D-Di	Diagnostica Stago	Quantitative	Latex-enhanced turbidimetric immunoassay	FEU	0.5 µg/mL	Exclusion
Central lab	Tina-Quant D-Dimer	Roche Diagnostics	Quantitative	Latex-enhanced turbidimetric immunoassay	FEU	0.5 µg/mL	Exclusion
Central lab (or POC)	VIDAS D-Dimer	bioMérieux	Quantitative	Enzyme-linked fluorescent assay	FEU	500 µg/L	Exclusion
POC	AQT90 FLEX D-dimer	Radiometer Medical ApS	Quantitative	Time-resolved fluorometry	NA	500 µg/L	NA
POC	Clearview Simplify	Agen Biomedical	Qualitative	Solid-phase immunochromatography	Neg/pos	80 µg/L	NA
POC	Pathfast D-Dimer	Mitsubishi Kagaku Iatron	Quantitative	Chemiluminescent immunoassay	FEU	0.686 µg/mL	NA
POC	Roche Cardiac D-dimer	Roche Diagnostics	Quantitative	Solid-phase immunochromatography	FEU	0.5 µg/mL	NA
POC	Stratus CS Acute Care D-dimer	Siemens Healthcare Diagnostics	Quantitative	Fluorescent immunoassay	FEU	450 µg/L	Exclusion
POC	Triage D-dimer	Biosite Diagnostics	Quantitative	Fluorescent immunoassay	DDU	350 µg/L	Aid in diagnosis

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
