# Peer review of "D-dimer Testing in Pulmonary Embolism with a Focus on Potential Pitfalls: A Narrative Review"

_diagnostics, 2022, doi:10.3390/diagnostics12112770_

Round 1
Reviewer 1 Report
1. The English need improvement since there are some grammatical and syntax errors in the manuscript. For example,
· in line number 42, the words “on D-dimer” may be as “of D-dimer”;
· in line number 57, “formation” as “the formation”;
· in line number 70, “physiological” as “the physiological”;
· in line number 94 and 96, “lower” as “a lower”;
· in line number 99, “performance” as “the performance”;
· in line number 102, “a lower” as “lower”;
· in line number 128, “of 50” as “50”;
· in line number 130, “was” as “were”;
· in line number 148, “A first” as “The first”;
· in line number 182, “anticoagulant” as “the anticoagulant”;
· in line number 188, “vertical” as “a vertical”;
· in line number 190, “low” as “the low”;
· in line number 203, “temperature” as “the temperature”;
· in line number 205, “analysis” as “to analysis”;
· in line number 256, “was” as “were”;
· in line number 288, “existence” as “the existence”;
· in line number 308, “comparison” as “a comparison”;
· in line number 314, “precision” as “a precision”;
· in line number 340, “clinical” as “the clinical”;
· in line number 352, “to use” as “using”.
The grammar mistakes which are not mentioned here are also to be checked and corrected properly.
2. There are some typing mistakes as well, and authors are advised to carefully proof-read the text. For example,
in line number 42, 319, and 322, the words “postanalytical” may be as “post-analytical or post analytical”;
· in line number 43, “discussing” as “discuss”;
· in line number 77, “Interest of” as “Interest in”;
· in line number 86, “laboratory developed” as “laboratory-developed”;
· in line number 104, “of validate” as “of validated”;
· in line number 125, “rule-out” as “rule out”;
· in line number 135, “ruling-out” as “ruling out”;
· in line number 140, “by as” as “as”;
· in line number 214, “mostly” as “most”;
· in line number 230, “impact of” as “impact on”;
· in line number 310, “cutoffs,” as “cutoffs”.
The typos not mentioned here are also to be checked and corrected properly.
3. Check the abbreviations throughout the manuscript and introduce the abbreviation when the full word appears the first time in the abstract and the rest part of the text and then use only the abbreviation (For example, clinical decision rule (CDR), fibrinogen equivalent unit (FEU) etc.,). Make a word abbreviated in the article that is repeated at least three times in the text, not all words need to be abbreviated. The authors should avoid using abbreviation in the keywords.
4. The literature search should be described in detail. The authors are encouraged to include the database, search engines (like PubMed, ScienceDirect, Google scholar etc.,), the keywords used etc., which may be included in the introduction section or separately in the materials heading since it is a review article.
5. How many articles obtained from each of the search engines? What is the inclusion and exclusion criteria? What is the type of article included in this manuscript? How many articles are included for this manuscript? A diagram depicted the literature search should be included (If possible only) for better understanding and outcome.
6. The introduction part appears less informative about the pulmonary embolism, thus this section should be indicated as detailed to understand the manuscript in clear.
7. The technical terms (Latin Phrase) “in vitro (in line number 100)” should be italic and it should be checked all over the manuscript.
8. The table legends should be improved and a proper footnote should be given. All legends should have enough description for a reader to understand the figure without having to refer back to the main text of the manuscript. For example, the necessary expansion may be given for abbreviations used.
9. In the conclusion, the authors may be included the limitation of the present findings and future direction for a better understanding of the manuscript.
10. The reference are not cited properly it should be checked and corrected properly. For example,few reference full name of the journals has been given and for others only short form has been given. It should be carefully checked and corrected as per the journal format.
Author Response
1. The English need improvement since there are some grammatical and syntax errors in the manuscript. For example, · in line number 42, the words “on D-dimer” may be as “of D-dimer”; · in line number 57, “formation” as “the formation”; · in line number 70, “physiological” as “the physiological”; · in line number 94 and 96, “lower” as “a lower”; · in line number 99, “performance” as “the performance”; · in line number 102, “a lower” as “lower”; · in line number 128, “of 50” as “50”; · in line number 130, “was” as “were”; · in line number 148, “A first” as “The first”; · in line number 182, “anticoagulant” as “the anticoagulant”; · in line number 188, “vertical” as “a vertical”; · in line number 190, “low” as “the low”; · in line number 203, “temperature” as “the temperature”; · in line number 205, “analysis” as “to analysis”; · in line number 256, “was” as “were”; · in line number 288, “existence” as “the existence”; · in line number 308, “comparison” as “a comparison”; · in line number 314, “precision” as “a precision”; · in line number 340, “clinical” as “the clinical”; · in line number 352, “to use” as “using”. The grammar mistakes which are not mentioned here are also to be checked and corrected properly. The grammar mistakes cited by the reviewer were reviewed and corrected accordingly. The manuscript was carefully checked for additional grammar mistakes. 2. There are some typing mistakes as well, and authors are advised to carefully proof-read the text. For example, in line number 42, 319, and 322, the words “postanalytical” may be as “post-analytical or post analytical”; · in line number 43, “discussing” as “discuss”; · in line number 77, “Interest of” as “Interest in”; · in line number 86, “laboratory developed” as “laboratory-developed”; · in line number 104, “of validate” as “of validated”; · in line number 125, “rule-out” as “rule out”; · in line number 135, “ruling-out” as “ruling out”; · in line number 140, “by as” as “as”; · in line number 214, “mostly” as “most”; · in line number 230, “impact of” as “impact on”; · in line number 310, “cutoffs,” as “cutoffs”. The typos not mentioned here are also to be checked and corrected properly. The typing mistakes cited by the reviewer, for which we deeply apologize, were reviewed and corrected accordingly. The manuscript was carefully checked for additional typing mistakes. 3. Check the abbreviations throughout the manuscript and introduce the abbreviation when the full word appears the first time in the abstract and the rest part of the text and then use only the abbreviation (For example, clinical decision rule (CDR), fibrinogen equivalent unit (FEU) etc.,). Make a word abbreviated in the article that is repeated at least three times in the text, not all words need to be abbreviated. The authors should avoid using abbreviation in the keywords. The abbreviations were reviewed and adapted according to the reviewer’s comment. 4. The literature search should be described in detail. The authors are encouraged to include the database, search engines (like PubMed, ScienceDirect, Google scholar etc.,), the keywords used etc., which may be included in the introduction section or separately in the materials heading since it is a review article. The article was written as a narrative review. To make this point clear to the reader, the title of the manuscript was adapted accordingly, which is now “D-dimer testing in pulmonary embolism with a focus on potential pitfalls: a narrative review” The authors agree that a systematic review would be valuable but the topic would need to be drastically narrowed. A search on Pubmed on the last 10 years using the keywords “D-dimer AND pulmonary embolism” yielded 1,739 results, which illustrates the need for narrowing the subject. The interest of performing a systematic review on specific points was added to the conclusion in order to reflect this. 5. How many articles obtained from each of the search engines? What is the inclusion and exclusion criteria? What is the type of article included in this manuscript? How many articles are included for this manuscript? A diagram depicted the literature search should be included (If possible only) for better understanding and outcome. Please see answer to comment # 4. 6. The introduction part appears less informative about the pulmonary embolism, thus this section should be indicated as detailed to understand the manuscript in clear. Information on pulmonary embolism was added in the introduction for clarity’s sake as this was indeed missing. 7. The technical terms (Latin Phrase) “in vitro (in line number 100)” should be italic and it should be checked all over the manuscript. Thank for making the point. This was corrected accordingly. 8. The table legends should be improved and a proper footnote should be given. All legends should have enough description for a reader to understand the figure without having to refer back to the main text of the manuscript. For example, the necessary expansion may be given for abbreviations used. The legend on Table 1 was adapted accordingly and in our opinion now better describes the content of the table. All abbreviation were properly cited. 9. In the conclusion, the authors may be included the limitation of the present findings and future direction for a better understanding of the manuscript. The conclusion was adapted accordingly, and the interest of systematic reviews was suggested as well as future issues caused by upcoming regulations. 10. The reference are not cited properly it should be checked and corrected properly. For example, few reference full name of the journals has been given and for others only short form has been given. It should be carefully checked and corrected as per the journal format. The journal’s recommendations were followed and the “MDPI ACS Journals” output style for Endnote was applied. The references were carefully checked for errors. DOI were added whenever available.
Reviewer 2 Report
This is a review article, and it is not systematic, meaning that it is prone to a variable degree of bias. It would have been much better if the review was systematic, but the scope would have to be narrowed then. Secondly, it would be great to add high-quality primary studies to resolve some problems or suggest which ones seem to be missing the most to steer future primary studies. Thirdly, and most interestingly, you should demonstrate if there are studies that question the use of the D-dimer in specific situations, which would give this article a more operative meaning and function. Are there negative studies, disproving its use, did you manage to identify and read those and provide some kind of response to those? This is why I asked for a systematic review since it aims to identify not just the positive, but also negative sides, which are difficult to identify in the basic search and reading of the published articles. Are there agreement studies between the measurements of various suppliers/methods? This would be an interesting addition to the paper.
Author Response
1. This is a review article, and it is not systematic, meaning that it is prone to a variable degree of bias. It would have been much better if the review was systematic, but the scope would have to be narrowed then. The authors would like to thank reviewer 2 for this comment and agree that a systematic review would be a valuable addition to the current literature, although the topic of the present manuscript is too broad to be treated in such a systematic manner (see also the authors’ responses to reviewer 1’s comments #4 and #5). The narrative nature of this review was added to the title for transparency purposes, which now is “D-dimer testing in pulmonary embolism with a focus on potential pitfalls: a narrative review”. 2. Secondly, it would be great to add high-quality primary studies to resolve some problems or suggest which ones seem to be missing the most to steer future primary studies. As answered to comment #1, the authors have carried out a narrative review, including the articles that they believed more useful. The main challenge remains the standardization of assays (form preanalytical to reporting), which could enable us to move away from the postulate that each lab needs to validate their threshold and reach interoperability between assays through calibration curves or else, to implement newly proposed age- or pretest-adjusted thresholds. 3. Thirdly, and most interestingly, you should demonstrate if there are studies that question the use of the D-dimer in specific situations, which would give this article a more operative meaning and function. Are there negative studies, disproving its use, did you manage to identify and read those and provide some kind of response to those? This is why I asked for a systematic review since it aims to identify not just the positive, but also negative sides, which are difficult to identify in the basic search and reading of the published articles. A sentence was added to draw attention to the fact that D-dimer may also be used in specific populations. This was however not further developed. Our research did not lead us to find negative studies on the use of D-dimer, and its use along with clinical decision rules is endorsed by various (inter)national organizations. 4. Are there agreement studies between the measurements of various suppliers/methods? This would be an interesting addition to the paper. This is a crucial issue indeed. In a post-hoc analysis of the YEARS study, Hamer et al. compared the performance of four D-dimer assays for the detection of PE (i.e., Liatest, Innovance, Tinaquant, Vidas). They confirmed that D-dimer assays differed significantly and that the impact on follow-up imaging did exist. Nevertheless, a NPV of at least 97% was shown for all assays (Hamer 2021 Thrombosis Research). Very high NPV were also found in the study of Djurabi et al. using the Tinaquant and Vidas D-dimer assays (99.4 and 99.7%, respectively) (Djurabi 2009 Thrombosis Research), as well as in the ADJUST-PE study but not with the same patients (Righini 2014 JAMA).
Round 2
Reviewer 1 Report
1. There are some grammatical, alignment and typographical errors are noted in the manuscript and it should be thoroughly checked and corrected throughout the manuscript.
For example,
· all over the manuscript, the word “postanalytical” may be as “post-analytical”;
· in line number 72, “the circulation” as “circulation”;
· in line number 95, “well established” as “well-established”;
· in line number 96, “relevant” as “the relevant”;
· in line number 102, “lower” as “a lower”;
· in line number 195, “blood” as “the blood”.
2. The authors have not carried out the abbreviations properly. In line number both abbarviaion and expansion has given for “clinical decision rule (CDR)”, but in line number 370, again the authors have given only expansion and it should be replaced as “CDR”. The same may be checked all other abbreviations used in the manuscript.
Author Response
Thank you for pointing out these mistakes. The mistakes cited were reviewed and corrected accordingly. The term “preanalytical” was also replaced by the term “pre-analytical”. The manuscript was carefully checked for additional mistakes.
Reviewer 2 Report
The manuscript was improved in line with the suggestions
Author Response
Thank you